# Soil fungal networks maintain local dominance of ectomycorrhizal trees

Minxia Liang[1], David Johnson [2], David F. R. P. Burslem [3], Shixiao Yu [1], Miao Fang[1], Joe D. Taylor[4], Andy F. S. Taylor[3,5], Thorunn Helgason [6] & Xubing Liu [1✉]

The mechanisms regulating community composition and local dominance of trees in species-rich forests are poorly resolved, but the importance of interactions with soil microbes is increasingly acknowledged. Here, we show that tree seedlings that interact via root-associated fungal hyphae with soils beneath neighbouring adult trees grow faster and have greater survival than seedlings that are isolated from external fungal mycelia, but these effects are observed for species possessing ectomycorrhizas (ECM) and not arbuscular mycorrhizal (AM) fungi. Moreover, survival of naturally-regenerating AM seedlings over ten years is negatively related to the density of surrounding conspecific plants, while survival of ECM tree seedlings displays positive density dependence over this interval, and AM seedling roots contain greater abundance of pathogenic fungi than roots of ECM seedlings. Our findings show that neighbourhood interactions mediated by beneficial and pathogenic soil fungi regulate plant demography and community structure in hyperdiverse forests.

[1] Department of Ecology, School of Life Sciences and State Key Laboratory of Biocontrol, Sun Yat-sen University, Guangzhou 510275, China. [2] Department of Earth and Environmental Sciences, The University of Manchester, Manchester M13 9PT, UK. [3] School of Biological Sciences, University of Aberdeen, Aberdeen AB24 3UU, UK. [4] School of Environment and Life Sciences, University of Salford, Salford M5 4WT, UK. [5] The James Hutton Institute, Craigiebuckler, Aberdeen AB15 8QH, UK. [6] Department of Biology, University of York, Heslington, York YO10 5DD, UK. ✉email: liuxubing@mail.sysu.edu.cn

Investigating the mechanisms that regulate abundance and coexistence of different plant species in natural ecosystems is critical for understanding community structure and dynamics[1–3]. Many theories have been proposed to explain the maintenance of species richness in forests[1], such as competitive interactions, spatial or temporal resource partitioning, intermediate disturbance, density-dependent growth or mortality, and neutral community processes. In principle, these mechanisms all promote high alpha diversity. However, many tropical and subtropical forests are dominated in basal area and biomass terms by a small number of ectomycorrhizal (ECM) tree species in just a few plant families, which represent most of the canopy individuals[4]. In such forests, there may also be a high diversity of understory arbuscular mycorrhizal (AM) tree species from a greater number of families. However, the mechanisms that maintain differential abundance and diversity patterns between ECM and AM species remain poorly understood[5].

Interactions between plants and soil microbes are frequently shown to influence the maintenance of plant diversity[6–8]. Negative plant–soil feedback (PSF) effects mediated by soil-borne pathogens are widely detected in forest communities[6,9–11], which could cause disproportionately high mortality of conspecific[10,11] and closely related[12] seedlings at high density near their parent trees, making resources available for distantly related species that are resistant to those natural enemies, and helping to maintain high tree species diversity. However, the existence of monodominance or familial dominance by ECM species in many forests across the globe[13] challenges the wide body of evidence that negative PSFs are pervasive. High local dominance of one species or multiple species within one family suggests that negative PSFs are compensated by positive effects of high conspecific density. One possible advantage of high conspecific density is the opportunity to benefit from communities of mycorrhizal fungi that are associated with conspecific adults[5,14]. Mycorrhizal fungi provide host plants with nutrients and protection from antagonists in exchange for carbohydrates from photosynthesis[15], but the relative benefit of different types of mycorrhizal associations can be context dependent[7,16–18].

A critical mechanism by which new recruits could benefit from mycorrhizal fungi is through integration into 'common mycorrhizal networks';[19] such networks are formed when the mycelium of an individual mycorrhizal fungus connects multiple plants simultaneously, and in several ecosystems they have been shown to have key roles in nutrient transport[20,21], seedling establishment[22] and inter-plant signaling[23]. Frequently, many of the tree species that form low diversity patches or groves in otherwise high diversity forests form ectomycorrhizas[5,24]. If the ECM symbiosis is responsible for local dominance by ECM trees, then there must be a net benefit for recruitment close to conspecific adults, which is sufficient to offset the negative effects of high pathogen densities that might otherwise cause negative PSFs. Recent evidence suggests that ECM associations are linked to the expression of positive PSFs, while AM associations are linked to negative PSFs[7,17,25], but the mechanisms underpinning such contrasting effects of mycorrhizal type on community assembly are poorly resolved. Here, we test the specific role of common mycorrhizal networks in regulating PSFs, and quantify how access to root-associated fungal networks affects growth, survival and pathogen loads of seedlings in patches of subtropical forest dominated by adult trees that associate with either ECM or AM fungi. We find that the survival and fitness of ECM tree species, which become canopy-dominants in hyper-diverse tropical and subtropical forests, increase markedly when seedlings could connect to fungal networks associated with neighbouring adult trees. By contrast, fungal networks do not promote survival of seedlings with arbuscular mycorrhizas, and these species are usually restricted to the understory.

## Results

**Long-term field survey on seedling dynamics.** To test whether the density-dependent effects on seedling recruitment differ between ECM and AM trees, we established 1200 1-m² seedling quadrats which were regularly distributed within six, 1-ha permanent plots in a subtropical forest at the Heishiding Nature Reserve of south China[12], and monitored survival of all woody plant seedlings within these quadrats every spring from 2008 to 2017. We recorded 17,824 individuals belonging to 130 species, 82 genera and 48 families over the 10 years. The first-year survival of newly germinated seedlings was significantly and positively associated with the conspecific seedling density for ECM tree species but not for AM species (Fig. 1). Additionally, ECM seedling survival was significantly promoted when present in neighbourhoods with a greater basal area of conspecific adults, but the survival of AM seedlings was significantly inhibited by increasing local conspecific adult basal area (Fig. 1). These results suggest that ECM plant species experienced overall positive density-dependent effects, while recruitment of AM plant species sustained negative density dependence[25], which is consistent with recent experimental evidence from some temperate and Mediterranean communities[7,17].

**Impacts of soil fungal networks on seedling demography.** We conducted two hyphal exclusion experiments with in-growth cores, to explicitly determine whether soil biota mediated the conspecific density-dependent effects on seedling survival and growth, and to test how connection to fungal networks beneath adult trees influences the direction and strength of PSFs. We selected five common ECM plant species and five common AM species as experimental units, and chose one location for each species where it was locally monodominant. Mesh-walled in-growth cores possessing 35 μm (which prevents access to roots but not mycorrhizal hyphae) or 0.5 μm (which excludes both) nylon mesh on the bottom and covering windows along the sides[26] were inserted to 28 cm depth in the soil and then back-filled with soil from each site, thus ensuring seedlings

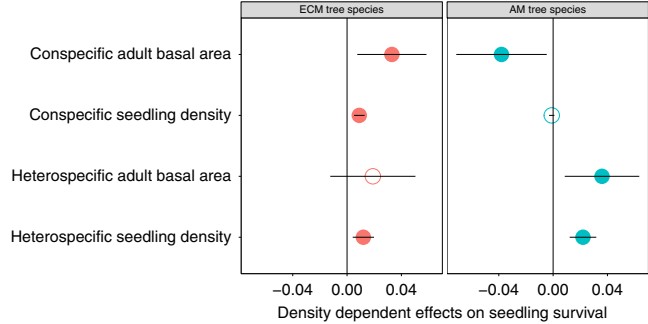

**Fig. 1 Density-dependent effects on first-year survival for ectomycorrhizal (ECM) and arbuscular mycorrhizal (AM) tree seedlings.** Left panel: ECM tree species. Right panel: AM tree species. For all of the 17,824 seedlings that were newly germinated during 2009–2016, we analysed their survival over the first year using a generalized linear mixed-effects model, in which we entered conspecific seedling density, heterospecific seedling density, conspecific adult density and heterospecific adult density as fixed factors, and species, recruitment year and location of each seedling as random factors, assuming a logit-link function and binomial error. Values represent regression coefficients and associated 95% confidence intervals. Bold solid circles indicate significant at $P < 0.05$, and hollow circles indicate non-significant differences with zero.

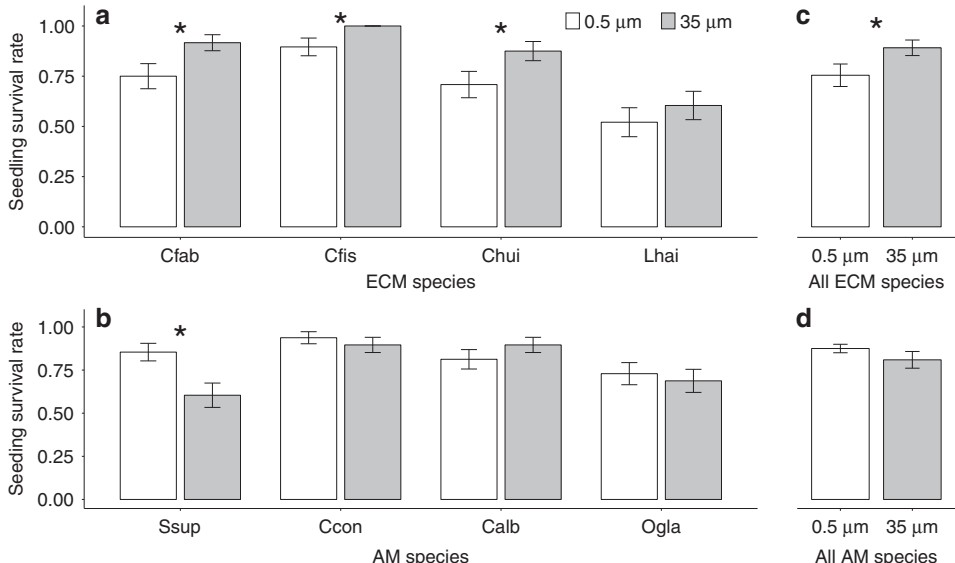

**Fig. 2 Effect of mycorrhizal hyphal connections on seedling survival for ECM and AM tree species.** Values represent seedling mean survival rates ± s.e.m. for four months after transplanting in the survival experiment. Species studied were: **a** ECM trees, including *Castanopsis fabri* (Cfab), *Castanopsis fissa* (Cfis), *Cyclobalanopsis hui* (Chui), *Lithocarpus haipinii* (Lhai); **b** AM tress, including *Schima superba* (Ssup), *Cryptocarya concinna* (Ccon), *Canarium album* (Calb) and *Ormosia glaberrima* (Ogla). Asterisks indicate significant differences at $P < 0.05$ based on one-sided two-proportion $z$-tests ($n = 48$ biologically independent seedlings for each treatment). Source data are provided in a Source data file.

transplanted into the cores were exposed to similar initial densities of soil hyphae.

For the seedling survival experiment, newly germinated seedlings lacking mycorrhizal colonization of four ECM and four AM species, were transplanted into in-growth cores at sites that were dominated by their conspecific adults. We found that seedlings of all ECM species had significantly greater survival after four months in 35 μm than 0.5 μm mesh cores (Fig. 2a, c), but survival of most AM species showed no differences between the two mesh size treatments (Fig. 2b, d, Supplementary Table 1). The significant promotion of ECM seedling survival by access to fungal hyphal connections at sites dominated by ECM adult trees (Fig. 2c, Supplementary Table 1) suggests that the ECM symbiosis counteracts the negative effects of pathogenic fungi in soils dominated by conspecific adults, and also that ECM trees generally tend to be more dependent on mycorrhizal fungi and benefit more from the symbioses than AM plants. Such effects may occur because of direct competition for plant carbon, sites of colonization on roots, enhanced plant nutrition, and induced plant resistance[16]. The absence of a significant effect of mesh size on seedling survival of AM species (Fig. 2d) suggests that AM fungi do not confer an equivalent survival advantage to that provided by ECM fungi[27].

In a second experiment quantifying seedling growth, 1-year old seedlings of three ECM and three AM focal species were transplanted into in-growth cores at six sites dominated in each case by conspecific adults of one of the six focal species using a fully reciprocal design. After nine months, mean seedling growth of all species was significantly promoted in 35 μm mesh cores when hyphae of seedlings could interact with fungal networks of surrounding adult trees compared to the 0.5 μm mesh treatment where hyphal access was prevented ($P = 0.002$, Supplementary Table 2). These positive biomass responses were seen in five of the six focal species at sites dominated by their conspecific adults (Fig. 3, red bars). When seedlings were grown at sites dominated by species of the same mycorrhizal type (ECM vs. AM fungi), the effect of allowing hyphal passage was always positive on seedling

growth although not always significant (Fig. 3). We detected a significant interaction effect between seedling mycorrhizal type and site mycorrhizal type on seedling biomass in the model that best fit the data (Supplementary Table 2), suggesting that tree species with the same mycorrhizal type may share connections to a common fungal network. There were no coincident effects of the hyphal connection treatment on seedling growth of AM species at ECM dominated sites (Fig. 3, light blue bars in the left panels) or ECM species at AM sites (Fig. 3, pink bars in the right panels). However, there were significantly positive effects of hyphal connections on *Castanopsis fissa* seedling biomass at the *C. fabri* dominated site, and on *C. fabri* biomass at the *C. fissa* site (both ECM species; Fig. 3), suggesting that closely related species may share the beneficial effects of hyphal connections. A significant interaction effect between seedling mycorrhizal type and site mycorrhizal type was also detected on root mycorrhizal colonization rates (Supplementary Table 3). Moreover, mycorrhizal colonization was significantly greater in cores possessing 35 μm mesh than 0.5 μm mesh (Supplementary Table 3), indicating that the mesh treatments were effective at controlling dispersal and colonization of mycorrhizal fungi in the cores. Root colonization by mycorrhizal fungi of seedlings in 35 μm mesh cores was consistently greater in conspecific vs. heterospecific environments (Supplementary Table 3), suggesting that growth responses in 35 μm mesh cores were not simply a consequence of seedlings having greater capacity to explore more soil through production of their own hyphae, but were in fact driven by integration into root-associated fungal networks with neighbouring adult trees. Meanwhile, moisture content did not differ between core types (Supplementary Fig. 1).

When comparing seedling biomass at the sites dominated by conspecific vs. heterospecific adults, all ECM species had significantly greater biomass at the home site than at the away sites in the 35 μm mesh treatment, but not in the 0.5 μm mesh treatment (Fig. 4, white bars; Supplementary Fig. 2, left panels). Conversely, the biomass of AM species was significantly inhibited at conspecific relative to heterospecific sites for both mesh

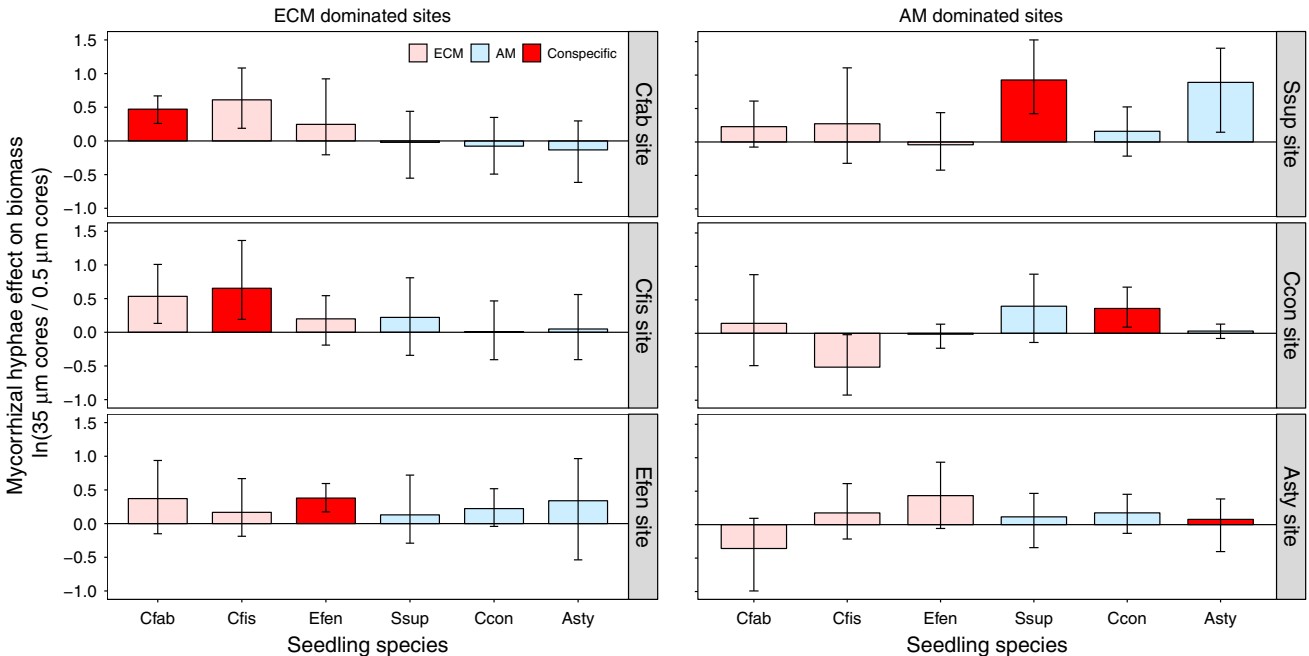

**Fig. 3 Effects of mycorrhizal hyphal connections on seedling growth at ECM or AM dominated sites.** Left panels: ECM dominated sites. Right panels: AM dominated species. Pink: ECM tree species; Light blue: AM tree species; Red: tree species at sites dominated by conspecific adults. Species studied were: *Castanopsis fabri* (Cfab), *Castanopsis fissa* (Cfis), *Engelhardia fenzelii* (Efen), *Schima superba* (Ssup), *Cryptocarya concinna* (Ccon), and *Artocarpus styracifolius* (Asty). Values represent log response ratios and associated 95% confidence intervals, which compared seedling mean biomass of in-growth cores with 35 μm vs. 0.5 μm mesh in the growth experiment ($n = 6$ biologically independent seedlings for each treatment).

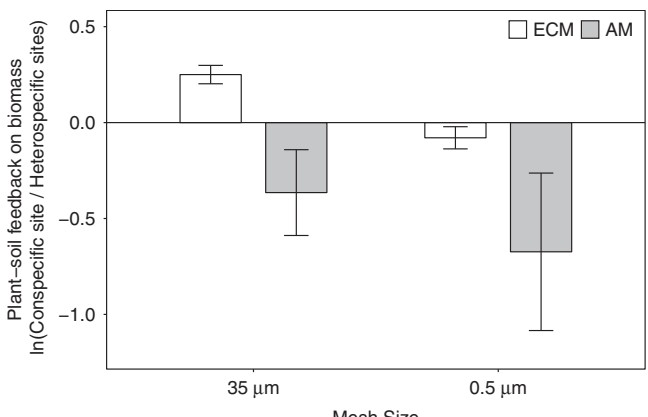

**Fig. 4 Overall plant–soil feedbacks on seedling growth for ECM and AM species.** White: ECM tree species; grey: AM tree species. Values represent mean log response ratios ± s.e.m. ($n = 3$ independent focal species for each mycorrhizal type), which compared seedling mean biomass of in-growth cores at conspecific ($n = 6$ biologically independent seedlings) vs. heterospecific ($n = 30$ biologically independent seedlings) sites. Source data are provided in a Source data file.

treatments (Fig. 4; Supplementary Fig. 2, right panels), and the negative effects were less in 0.5 μm than 35 μm cores (Fig. 4, grey bars).

**Molecular analysis on root-associated fungi**. We conducted molecular analysis on fungi associated with the roots of naturally established seedlings of all ten species from the same forest, to compare the composition and relative abundance of pathogenic fungi between ECM and AM plant species. We found that species richness of putative pathogens on fine roots did not differ significantly between ECM and AM trees (mean richness ± SE was

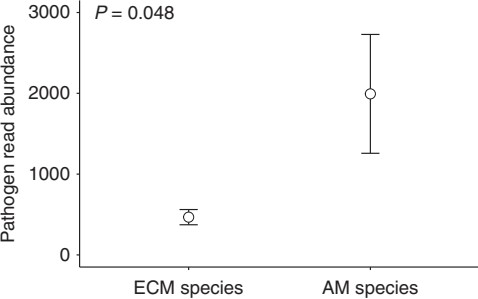

**Fig. 5 Abundances of pathogenic fungi associated with ECM and AM seedling roots.** Values represent mean read numbers of pathogenic OTUs associated with fine roots of field seedlings, and error bars represent standard errors ($n = 30$ biologically independent fine root samples). $P = 0.048$ based on independent two-sample *t*-test. Source data are provided in a Source data file.

$11.03 ± 0.62$ and $11.83 ± 0.64$, respectively), but that the read abundance of pathogenic fungi was significantly less on roots of ECM plants than AM plants (Fig. 5). AM seedling roots were associated with greater abundances of putative pathogenic fungi than ECM roots, suggesting that ECM fungi were more capable of limiting the abundance of root pathogens than AM fungi, probably by depressed accumulation rates of fungal pathogens around seedling roots[14].

## Discussion

Our results provide clear evidence that seedling survival and growth were influenced when they had the potential to be connected via root-associated fungal networks to neighbouring adult trees, and offer a mechanistic explanation for how different types of mycorrhizal hyphal connections alter the magnitude and direction of PSFs. Previous studies have shown how interactions between plants and their associated soil biota, particularly soil-

borne pathogens, drive plant community dynamics and maintain high diversity[6,9–11,16]. Mycorrhizal fungi were also found to promote species coexistence by regulating soil phosphorus partitioning among hosts with different mycorrhizal types in tropical[18,28] and temperate[29,30] ecosystems. However, it has yet to be determined how the different diversity patterns of ECM and AM tree species are maintained in hyperdiverse forests[5]. Our study addresses this question by showing that accessibility to seedlings of established host-specific ECM fungal networks results in positive PSFs, which may contribute to the local dominance of ECM trees. By contrast, the high diversity of understory AM species is maintained due to the overall negative PSFs especially in sites dominated by conspecific adults.

The positive effects on seedling biomass of access to mycorrhizal hyphae supported by conspecific trees indicate that fungal networks promote seedling nutrition and growth for all ECM and some AM species (Fig. 3); however, we do not know the precise mechanisms regulating such effects. It is possible that access to soils with greater density and larger networks of compatible hyphae, due to the presence of adult plants, could lead to the more efficient acquisition of soil resources by seedlings. Alternatively, seedling performance could be mediated by inter-plant transfer of materials if seedlings formed common mycorrhizal networks with adult plants. For example, mycorrhizal symbiosis can facilitate inter-plant transfer of both defence signals[23] and carbon[20,21]. Direct competition for sites of colonisation between mycorrhizal and pathogenic fungi is another important mechanism that may change the directionality of PSFs between ECM and AM trees, which is supported by the significantly different abundances of pathogenic and mycorrhizal fungi associated with ECM and AM seedling roots (Fig. 5). Unlike AM fungi, ECM fungi physically provide more effective protection to woody plants against root pathogens[7,25,31], which could have important impacts on species coexistence and local community structure of hyperdiverse forests[16].

The significant mycorrhizal facilitation and pathogenic inhibition effects of home vs. away sites on seedling growth (Fig. 4) indicated that both positive and negative PSFs were host specific. Although mycorrhizal fungi have traditionally been considered to have relatively lower host specificity than fungal pathogens[8,32], the full reciprocal design of our seedling growth experiment revealed that the beneficial effects of neighbouring adults through mycorrhizal connections were specific at the species or genus level (Fig. 3), which corresponds to the emerging view that host specificity is higher than previously thought[33–36].

While negative PSFs caused by soil-borne pathogens have been shown to be a key driver of plant diversity in forest communities[6,10,11], our study suggests that tree community structure in hyperdiverse forests cannot be explained solely by inhibitory effects of pathogens but also involves mycorrhizal facilitation effects. Adult ECM trees support mycorrhizal hyphal connections that enhance seedling growth and survival and inhibit antagonists, which may ultimately result in canopy dominance of ECM tree species. By contrast, AM seedlings experience conspecific inhibition because AM fungi are less effective at protecting seedlings against host-specific root pathogens[16], and the overall negative density-dependent effects of AM trees contributes to the high diversity of AM species in species-rich forests. By regulating the growth of ECM seedlings and influencing the efficacy of root pathogens across species, ECM fungi have critical roles in shaping community structure and dynamics in subtropical forests, and may be a critical factor explaining global patterns of ECM and AM trees in forests[13].

## Methods

**Study site**. Our field site is located at the Heishiding Nature Reserve (111°53′E, 23° 27′N, 150–927 m a.s.l.) in Guangdong Province of south China, which consists of 4200 ha of subtropical evergreen broad-leaved forest. The region is located on the Tropic of Cancer and has a subtropical moist monsoon climate. Mean annual temperature is 19.6 °C and mean monthly temperatures range from 10.6 °C in January to 28.4 °C in July. Annual precipitation is about 1744 mm, occurring mainly between April and September (79% of annual rainfall), and a pronounced dry season lasts from October to March.

**Density-dependent effects for ECM and AM seedlings**. To investigate the effects of neighbouring adults and seedlings on performance at the seedling stage, in spring 2008 we demarcated 1200 1 × 1 m quadrats, which were regularly spaced within six 1-ha permanent plots at the field site[12]. We tagged all tree individuals with a diameter at breast height (DBH) ≥ 1 cm in the 1-ha plots. We identified their species, mapped their locations, and measured their DBH. In the seedling quadrats, we surveyed all seedlings of woody plants (from new emergents to DBH < 1 cm) every spring from 2008 to 2017. At each census, we tagged new seedlings, determined their species, and measured their heights.

**In-growth core experiments**. To test whether seedlings benefit from mycorrhizal hyphal connections to neighbouring plants, we conducted two independent hyphal exclusion experiments with mesh cores to evaluate the external mycorrhizal mycelium effects on seedling survival and growth respectively[37]. We selected ten common tree species in the study area as focal species, including five ECM species: *Castanopsis fabri* (Fagaceae), *Castanopsis fissa* (Fagaceae), *Cyclobalanopsis hui* (Fagaceae), *Lithocarpus haipinii* (Fagaceae), and *Engelhardia fenzelii* (Juglandaceae), and five AM species: *Schima superba* (Theaceae), *Cryptocarya concinna* (Lauraceae), *Canarium album* (Burseraceae), *Ormosia glaberrima* (Fabaceae), and *Artocarpus styracifolius* (Moraceae). The mycorrhizal status of each species was determined by ref. [38]. These species were selected based on their abundance and seed availability at the time of collection, and the phylogenetic imbalance between ECM and AM species objectively represented the community composition at the field site, i.e. AM are widespread among tree species and ECM are restricted to fewer tree taxa. The seedling growth experiment was conducted between March 2017 and January 2018 with six tree species, and the survival experiment was conducted from March to August 2018 with eight species (Supplementary Table 4). For each species, we collected fruits and seeds throughout the study site during autumn and winter of 2015 (the growth experiment) and 2017 (the survival experiment). Seeds were surface-sterilized (1 min 70% ethanol, 3 min 2.63% NaOCl, 1 min 70% ethanol, 1 min distilled water) and kept in a refrigerator at 4 °C until late March of the next year. Seeds were left to germinate in plastic boxes filled with an autoclaved mixture of soil and sand. Newly germinated seedlings were transplanted into the mesh cores for the survival experiment, but seedlings were regularly watered and cultured for one year before being used for the growth experiment, to maximize competition among seedling individuals.

Mesh-walled cores were assembled from 16 cm diameter × 30 cm deep PVC piping, perforated with six 8-cm-diameter windows which were regularly distributed along the side with three of them at the depth of 4–12 cm and the other three at 16–24 cm. The cores were lined with 35 μm or 0.5 μm nylon mesh (Plastok Associates Ltd, Birkenhead, UK) to cover the bottom and the windows, which was attached using transparent superglue (Pattex®, Henkel Adhesives Ltd., Shantou, China). Nylon mesh with a pore size of 35 μm excludes roots of neighbouring plants, but allows mycorrhizal hyphae access to the transplanted seedlings; while cores with a 0.5 μm mesh exclude both fine roots and hyphae, with only free-living soil microorganisms passing through[27,39]. We then covered the sides and the bottoms of all cores with 2 mm nylon mesh, to prevent soil fauna damaging the smaller size meshes. The mesh treatments may cause potential underestimation of negative soil effects as they would exclude dispersal of root-feeding nematodes, but these should have the same impact on seedlings of both mesh sizes.

For each focal species, we selected one 30 × 30 m site where the DBH-weighted relative abundance of the focal species was greater than 30% among all tree individuals with diameter at breast height (DBH) ≥ 5 cm, and there were no adult individuals of the other study species within each site. At each experimental site, 12 mesh cores were inserted to 28 cm depth in the soil, as this is the part of the soil profile where the majority of soil microbial activities take place[40]. The cores were randomly distributed within a 3 × 2 m area in the centre of each site, with a random distance of 30–50 cm between neighbouring cores. The soil was thoroughly mixed at each site and backfilled into the cores. We then let the cores stand for 4 weeks before seedling transplantation, to allow recovery from disturbance.

For the seedling survival experiment, seedlings of eight species were transplanted to the sites dominated by their conspecific adults, with 6 replicate cores for each mesh size and 8 seedlings per core. A total of 768 seedlings (n = 8 species × 2 mesh treatments × 6 core replicates × 8 seedlings per core) were transplanted at about one month after germination. For the seedling growth experiment, seedlings of six species were transplanted to each site dominated by a single focal species using a full reciprocal design. Treatments consisted of: (1) six focal species comprising three ECM and three AM species; (2) six corresponding sites dominated by one of the focal species; (3) two mesh sizes of cores (35 μm vs. 0.5 μm). Each focal species had one monoculture core with 6 conspecific seedlings for each mesh size treatment at each site. One-year old seedlings were used for this experiment, and we transplanted 432 seedlings in total (n = 6 sites × 6 species × 2 mesh sizes × 6 seedlings per core).

For both experiments, we randomly selected and transplanted the seedlings into the mesh cores. One week after the transfer of seedlings into cores, we removed the seedlings that were dead or poorly growing due to injuries during transplantation, and replaced them with new seedlings. All seedlings were allowed to grow for 4 months (the survival experiment) or 9 months (the growth experiment), and then harvested to determine their biomass. At the end of both experiments, we removed each core from the soil and then carefully removed the seedlings. Each seedling was washed to remove any attached soil and separated into shoot and root, and their dry weights measured after oven-drying at 60 °C for 72 h. At the harvest, we collected fresh root and soil samples from each core for laboratory analysis. We randomly collected 10 fine root fragments of 1 cm length from each seedling, and stored them in centrifuge tubes with a piece of wet filter paper in the bottom, kept at 4 °C and transferred to a laboratory within 2 days for analysis of mycorrhizal colonization. Mycorrhizal colonization of roots among focal species was quantified using the grid-line intersection method[18,41]. AM colonization was defined as typically aseptate hyphae, forming arbuscules and vesicles, and ECM colonization was quantified by counting the percentage of colonized root tips covered with fungal mantle and Hartig net.

To investigate the potential for water-logging inside the cores as a consequence of slow drainage through a small diameter mesh, we measured soil moisture content and temperature using the HydraProbe Sensors (Stevens Water Monitoring Systems Inc., Portland, USA) for each in-growth core. These measurements were repeated three times at the beginning, the middle and the end of the experiment, respectively. We found that soil moisture content and temperature did not differ significantly between 35 and 0.5 μm cores at all sites (Supplementary Fig. 1).

**Molecular analysis of root-associated fungi**. To assess the composition of pathogenic fungi associated with different tree species, we analysed the root-associated fungi for the ten focal species. For each species, six 5-g fine root samples were collected from randomly selected seedlings of the target species, which were naturally established within the six 1-ha permanent plots and had a similar size as the experimental seedlings. We surface sterilized each root sample with a 1-min wash in 99% ethanol, followed by a 6-min wash in 3.125% NaOCl, a 30-s wash in 99% ethanol, and a final rinse in sterile reverse osmosis-treated water. We extracted total genomic DNA from each sample using the CTAB (cetyl trimethylammonium bromide) protocol[42]. The nuclear ribosomal internal transcribed spacer region (ITS rDNA gene) was amplified by polymerase chain reaction (PCR) using the fungal primer set for ITS1-1F (5′-CTTGGTCATTTAGAGGAAGTAA-3′) and ITS2 (5′-GCTGCGTTCTTCATCGATGC-3′). The PCRs were performed in a 30 μL reaction mixture containing 3 μL of 2 μM primer, 10 μL of template DNA (1 ng/μL), 15 μL PhusionMasterMix 2× (Phusion® High-Fidelity PCR Master Mix with GC Buffer), and 2 μL of H$_2$O. The PCR amplification program was set as follows: samples were initially denatured at 98 °C for 1 min, 30 cycles of denaturation at 98 °C for 10 s, primer annealing at 50 °C for 30 s, and extension at 72 °C for 30 s. A final extension step of 5 min at 72 °C was added to ensure complete amplification of the target region. For each root sample, the PCR was conducted in triplicate and the PCR products were pooled to reduce PCR amplification biases. Then PCR products were purified with GeneJET DNA gel extraction kit (Thermo Scientific, USA) as directed by the manufacturer. The amplicon library was conducted with Ion Plus Fragment Library Kit 48 rxns and sequenced on an Ion S5™ XL SE600 next-generation sequencing system (Thermo Fisher Scientific Inc., UK).

We processed the raw sequence data using MOTHUR[43] (version 1.29). The pyrosequences were first denoised and low-quality reads with eight or more homopolymer bases, ambiguous base calls and/or average quality < 25 bases were identified and removed. Quality reads were subsequently assigned to each sample according to their unique barcodes. Chimeric sequences were identified and removed in each sample using UCHIME[44] (version 4.1), with the data sets themselves as the references. We obtained an average of 80,127 ± 302 (mean ± SE) clean reads for each root sample with a mean length of 247 ± 3 bp. Operational taxonomic units (OTUs) were identified at the 97% similarity level using UPARSE[45] (version 7.0.1001, http://drive5.com/uparse/). Using the BLAST taxonomy assignment method in QIIME (version 1.9.1, http://qiime.org/scripts/assign_taxonomy.html), we compared the OTU representative sequences against the UNITE ITS sequence database[46]. We assigned each identified fungal species to putative pathogen, mycorrhiza or saprotroph using the FunGuild algorithm[47] at the 'highly probable' and 'probable' confidence rankings. To investigate how pathogen resistance was affected by different mycorrhizal types, we normalized the OTU table to the minimum sequencing depth (78,998 clean reads) among different samples and calculated pathogen read abundance for ECM and AM trees, respectively, using reads of pathogenic OTUs as a relative measure of the abundance of pathogenic fungi for each root sample, as we found that the relative frequency of pathogens was significantly correlated with seedling performance[11].

**Statistical analysis**. For ECM trees and AM trees respectively, we constructed generalized linear mixed models[48] using the R package *lme4*[49], to test whether the first-year survival for newly germinated seedlings from 2008 to 2016 was related to conspecific adult density, conspecific seedling density, heterospecific adult density, and heterospecific seedling density, assuming a logit-link function and binomial error. For each focal seedling, we used the sum of conspecific or heterospecific

adult basal area within a radius of 10 m as a proxy for conspecific and heterospecific densities, respectively. Conspecific and heterospecific seedling densities were calculated as the conspecific and heterospecific seedlings within the same 1 × 1 m quadrat. We included recruitment year, location (quadrat and site), and species name of each seedling as random factors in the model to account for potential temporal, spatial, and interspecific variation.

We used two-proportion z-test to compare the seedling survival rates between different mesh size treatments for each species. To detect differences in seedling survival between mycorrhizal types and mesh sizes, we constructed generalized linear mixed-effects models[48] with a binomial error distribution and logit-link function, where mesh size, mycorrhizal type, and their interaction were fixed effects, and species, and core were treated as random effects in the models.

To quantify the mycorrhizal mycelium effects on seedling growth, we calculated log-response ratios comparing seedling total biomass between in-growth cores with 35 μm or 0.5 μm mesh for each focal species at each site. We estimated the 95% confidence interval of each log-response ratio by bootstrap resampling with 9999 repetitions. To investigate host specificity of PSF effects, we also compared seedling biomass between conspecific and heterospecific sites of each mesh size for each species, using pairwise t-tests. We then combined all ECM or all AM species together, and calculated log-response ratios comparing seedling growth at conspecific sites vs. heterospecific sites for each mesh size treatment, to reveal the overall response of ECM and AM species to the mycorrhizal hyphal connections at home or away soils. We also constructed linear mixed-effects models[48] to detect differences in seedling biomass between mycorrhizal types and mesh sizes, where mesh size, seedling mycorrhizal type, site mycorrhizal type, and the interaction between seedling and site types were fixed effects, and site, core, and species were treated as random effects in the models. Similar linear mixed-effects models were also used to reveal the effects of mesh size on root colonization rates. All statistical analyses were performed using R (version 3.5.1; R Core Team, Vienna, Austria).

**Reporting summary**. Further information on research design is available in the Nature Research Reporting Summary linked to this article.

## Data availability
Shade-house experimental data are available in the NERC Environmental Information Data Centre at https://doi.org/10.5285/f1d17e61-bb6c-47a9-a648-062c63ea7f16. Sequence data have been deposited in the NCBI Sequence Read Archive (SRA) under Project ID PRJNA627300. Field census data are available upon reasonable request from the ForestGEO data portal at http://ctfs.si.edu/datarequest/.

## Code availability
The R codes used in this study are available at https://github.com/liuxubing/common-mycorrhizal-networks.

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

## Acknowledgements

We are grateful to Weinan Ye, Dr Yinghua Luo, Dr Zongbo Peng, Jie Li, Dr Meng Xu, Fengmin Huang, Wenbin Li, Yongning Li, and also Dr Buhang Li and his field technician team for help with field data collection. We thank Yanwen Chen, Huidan Teng, and Dr Shan Luo for assistance in the field experiments. We acknowledge Dr. Fang Li for kindly providing the featured images of ECM fungi. This research was funded by the National Key Research and Development Program of China (Project No. 2017YFA0605100) and the National Natural Science Foundation of China (NSFC 31770466 to X.L. and 31870403 to M.L.), and partly supported by awards from the UK Natural Environment Research Council (NERC NE/M004848/1 and NE/R004986/1). D.J. is also supported by the N8 AgriFood programme.

## Author contributions

X.L., D.B., D.J., and M.L. designed the study, J.T., A.T., T.H., and S.Y. participated in the original idea generation. M.L., X.L., and S.Y. collected the long-term seedling demography data and performed molecular analysis on root-associated fungi. M.L. and M.F. conducted the seedling survival experiment, X.L., M.L., D.B., and D.J. conducted the growth experiment. M.L. and X.L. performed statistical analyses and wrote the paper, with substantial input from D.J. and D.B. on data interpretations, and all authors contributed to revisions.

## Competing interests

The authors declare no competing interests.
