## [Peer Review File · Nature Communications]

Editorial Note: This manuscript has been previously reviewed at another journal that is not operating a transparent peer review scheme. This document only contains reviewer comments and rebuttal letters for versions considered at Nature Communications .

Reviewers' Comments:

Reviewer #3:

Remarks to the Author:

4.5.20 Nature communications review

This study is a massive effort to examine the effects of common mycorrhizal networks in promoting ectomycorrhizal tree monodominance in the tropics. This is a really interesting question, that I think has puzzled many mycologists over the decades, and this paper does a great job showing data that EMF support positive density dependent growth and monodominance whereas AMF support negative density dependent growth. The effort involved in this paper is behemoth – at risk of offending the editors.... I can't see why it shouldn't go in Nature rather than Nature Communications. Usually I'm a pretty harsh reviewer but honestly I really enjoyed reading this paper.

Clearly I am not an original reviewer of the paper. However, reading through the responses to reviewer comments it appears that the authors took reviewer comments seriously and addressed all of them adequately. For the most part I agree with reviewer comments except for a few (Q5, Q7, Q30). For Q5 – I don't see what is wrong with backward model selection and I took several multivariate statistics classes as a graduate student. I also disagree with Q7. ITS works adequately for mycorrhizal fungi and any AMF it finds you can certainly infer are there. I would have no issue with you using ITS data to infer AMF presence and indeed would like to see that data in a supplemental table or figure. ITS might not get as many lineages as AMF specific primers, but if you see Lekberg and Opik "More bang for the buck? Can arbuscular mycorrhizal fungal communities be characterized adequately alongside other fungi using general fungal primers?" New Phyt 2019 they clearly show that ITS primers can assess many AM fungi although not quite the diversity that AMF specific primers do. I also disagree with Q30 – 4 weeks is ample time for many microbial communities to restore – certainly bacteria would recover in this time period. So I agree with reviewer response to this question. In general I liked this paper but just have a few minor comments below:

My own line comments:

Line 31: I believe the correct grammar here would be investigating mechanisms is critical

Lines 46-48: agree, this is a very interesting question!

Line 64-65: not that you need more citations, but a recent review paper Tedersoo et al Science 2020 "How mycorrhizal associations drive plant population and community biology" also talks about this very point at length and suggests that positive density dependence enhances EMF monodominance whereas negative density dependence causes diversity in AMF forests

Lies 68-69: I disagree with previous reviewer comments. I think common mycorrhizal networks is fine – just say CMNS proxied by access to root-associated fungal networks from different mesh sizes – because how else would you measure this??

Line 74: 1,200 1 m² seedling quadrats! What an amazing effort!

Line 77: >17k individuals!! That's very impressive. You must have had a lot of technicians/undergrad labor for this – I hope you acknowledge them in the acknowledgements

Lines 78-80: very interesting!

Line 86: and the speculations in Tedersoo et al Science 2020

Figure 2: legends are too small to read. I suggest you increase font size.

Lines 102-105: also that in general EMF trees tend to be more dependent on EMF and obligate on the symbioses than AMF plants.

Figure 3: font size should also be increased in legends/axes here.

Line 139: moisture content

Line 148: it would be interesting to at least include a supplementary table of the EMF and AMF species assessed here since I think ITS primers are valid and many readers would appreciate it as long as you have the data. Not everyone hates ITS primers like Reviewer 1 (clearly, seeing as ITS is the universal barcode of fungi – Schoch et al PNAS 2012)

line 186-187: Yes, the idea that EMF mantle better protects plant roots against pathogens than AMF was posited in the Tedersoo et al Science 2020 paper – I am not an author on that paper but I just read it last week and it discussed CMNs extensively so it is quite relevant here.

Line 195: also Gao et al Molecular Ecology 2013

Lines 372-374: this isn't really acceptable to me. I need to see accession numbers here. You can always set them to release upon publication but without an accession number it's not believable that you will do the work of uploading the data otherwise.

Reviewer #4:

Remarks to the Author:

This is a resubmission and I believe that Liang et al. satisfactorily addressed the reviewer's comments. While this is an observational study and doesn't provide direct causation, it does provide very compelling evidence that mycorrhizal networks are likely a possible mechanism in explaining forest species diversity. This is by no means the last study of this emerging paradigm and I hope this study will spur the scientific community to help advance this hypothesis into a possible theory.

Overall, this was a well-polished article and easily understandable. All my comments are minor, but I believe they will improve the readability.

Throughout the manuscript it is stated that high biodiversity is expected with AM-associated tree species due to a negative PSF (e.g. In 44 & 203). While references are provided, the mechanism is never really explained. Why would a high pathogen load can cause high diversity – what's the paradigm? You state "...which maintain high diversity by reducing the survivorship of conspecific seedlings in high density patches..." Based on this explanation, it still doesn't exclude the possibility of monodominance or a few genera taking over. It is just not obvious in the manuscript and I think 1-2 sentences can provide a fully-formed explanation.

Figures 2,3,4 & Sup 1: Please change the bar graphs into box plots to provide data transparency and it is more informative. Overall, bar graphs basically hide most of the results by just showing average and SE, at least with a box plot the reader can get a sense of how the data was distributed within the treatment.

Responses to the referee comments on NCOMMS-20-08509-T

We would like to thank the reviewers for their appreciation of our work and constructive comments, all the comments have been helpful and we have revised the manuscript based on these suggestions. The substance of our revision and our point-by-point responses to the reviewer comments (Reviewer 1: Q1-Q19; Reviewer 2: Q20-Q21) are detailed here.

REVIEWERS' COMMENTS:

Reviewer #3 (Remarks to the Author):

4.5.20 Nature communications review

This study is a massive effort to examine the effects of common mycorrhizal networks in promoting ectomycorrhizal tree monodominance in the tropics. This is a really interesting question, that I think has puzzled many mycologists over the decades, and this paper does a great job showing data that EMF support positive density dependent growth and monodominance whereas AMF support negative density dependent growth. The effort involved in this paper is behemoth – at risk of offending the editors.... I can't see why it shouldn't go in Nature rather than Nature Communications. Usually I'm a pretty harsh reviewer but honestly I really enjoyed reading this paper.

Clearly I am not an original reviewer of the paper. However, reading through the responses to reviewer comments it appears that the authors took reviewer comments seriously and addressed all of them adequately. For the most part I agree with reviewer comments except for a few (Q5, Q7, Q30).

Q1: For Q5 – I don't see what is wrong with backward model selection and I took several multivariate statistics classes as a graduate student.

Response: We agree with this comment that backward subtraction of terms would be useful for model selection. Given that we actually obtained the same results for both methods, we reported the main model results as suggested by Reviewer 1 in the final manuscript, to make the statistical approaches concordant between the two experiments.

Q2: I also disagree with Q7. ITS works adequately for mycorrhizal fungi and any AMF it finds you can certainly infer are there. I would have no issue with you using ITS data to infer AMF presence and indeed would like to see that data in a supplemental table or figure. ITS might not get as many lineages as AMF specific primers, but if you see Lekberg and Opik "More bang for the buck? Can arbuscular mycorrhizal fungal communities be characterized adequately alongside other fungi using general fungal primers?" New Phyt 2019 they clearly show that ITS primers can assess many AM fungi although not quite the diversity that AMF specific primers do.

Response: We agree with this comment that ITS primers can assess many AM fungi. As our main objective was to compare the differences between ECM and AM tree species, while ITS may have different sensitivity among different functional guilds, e.g. ITS1F may select against glomeromycota in favour of asco-basidiomycota, we only included the results for pathogenic fungi in our main text (Figure 5). We have provided the full table of functional guild identification on all fungal OTUs in the Source Data file (the sheet entitled “Fig 5”), including all pathogenic and mycorrhizal fungi as suggested by Q16. The raw sequencing data has also been deposited in the NCBI Sequence Read Archive (SRA) under Project ID PRJNA627300, and included as a supplementary information in the updated paper.

Q3: I also disagree with Q30 – 4 weeks is ample time for many microbial communities to restore – certainly bacteria would recover in this time period. So I agree with reviewer response to this question.

Response: We appreciate this agreement.

In general I liked this paper but just have a few minor comments below:

My own line comments:

Q4: Line 31: I believe the correct grammar here would be investigating mechanisms is critical

Response: We agree and have changed “are” to “is” on L31.

Q5: Lines 46-48: agree, this is a very interesting question!

Response: We appreciate this agreement.

Q6: Line 64-65: not that you need more citations, but a recent review paper Tedersoo et al Science 2020 “How mycorrhizal associations drive plant population and community biology” also talks about this very point at length and suggests that positive density dependence enhances EMF monodominance whereas negative density dependence causes diversity in AMF forests

Response: We agree and have added the Tedersoo et al Science 2020 paper to our reference list.

Q7: Lines 68-69: I disagree with previous reviewer comments. I think common mycorrhizal networks is fine – just say CMNS proxied by access to root-associated fungal networks from different mesh sizes – because how else would you measure this??

Response: We agree with this comment and have changed “the prediction that differential access to root-associated fungal networks regulates PSFs” to “the specific role of common mycorrhizal networks in regulating PSFs”.

Q8: Line 74: 1,200 1 m² seedling quadrats! What an amazing effort!

Response: We appreciate this plaudit.

Q9: Line 77: >17k individuals!! That's very impressive. You must have had a lot of technicians/undergrad labor for this – I hope you acknowledge them in the acknowledgements

Response: We agree and have acknowledged the field assistants as suggested: *“We are grateful to Weinan Ye, Dr Yinghua Luo, Dr Zongbo Peng, Jie Li, Dr Meng Xu, Fengmin Huang, Wenbin Li, Yongning Li, and also Dr Buhang Li and his field technician team for help with field data collection.”*

Q10: Lines 78-80: very interesting!

Response: We appreciate this plaudit.

Q11: Line 86: and the speculations in Tedersoo et al Science 2020

Response: We agree and have cited Tedersoo et al Science 2020 paper on L86.

Q12: Figure 2: legends are too small to read. I suggest you increase font size.

Response: We agree and have increased the legend font size of Figure 2.

Q13: Lines 102-105: also that in general EMF trees tend to be more dependent on EMF and obligate on the symbioses than AMF plants.

Response: We agree and have added this statement *“and also that ECM trees generally tend to be more dependent on mycorrhizal fungi and benefit more from the symbioses than AM plants”* **on L106.**

Q14: Figure 3: font size should also be increased in legends/axes here.

Response: We agree and have increased the font size in legends and axes of Figure 3.

Q15: Line 139: moisture content

Response: We agree and have changed “contents” to “content”.

Q16: Line 148: it would be interesting to at least include a supplementary table of the EMF and AMF species assessed here since I think ITS primers are valid and many readers would appreciate it as long as you have the data. Not everyone hates ITS primers like Reviewer 1 (clearly, seeing as ITS is the universal barcode of fungi – Schoch et al PNAS 2012)

Response: We have reported the results of pathogenic fungi (Figure 5) in our main text to ensure comparability between ECM and AM tree species, and have

provided the suggested table of the identified fungal OTUs in the Source Data file (“Fig 5” sheet), including all pathogenic and mycorrhizal fungal species that we identified. All of the raw sequencing data has also been deposited in the NCBI Sequence Read Archive (SRA) under Project ID PRJNA627300, and included as a supplementary information in the updated paper. Please refer to Q2 for more details.

Q17: line 186-187: Yes, the idea that EMF mantle better protects plant roots against pathogens than AMF was posited in the Tedersoo et al Science 2020 paper – I am not an author on that paper but I just read it last week and it discussed CMNs extensively so it is quite relevant here.

Response: We agree and have cited the Tedersoo et al 2020 paper on L187.

Q18: Line 195: also Gao et al Molecular Ecology 2013

Response: We agree and have added the suggested citation on L 195.

Gao, C. et al. Host plant genus-level diversity is the best predictor of ectomycorrhizal fungal diversity in a Chinese subtropical forest. *Mol Ecol.* **22**, 3403-3414 (2013).

Q19: Lines 372-374: this isn’t really acceptable to me. I need to see accession numbers here. You can always set them to release upon publication but without an accession number it’s not believable that you will do the work of uploading the data otherwise.

Response: We agree with this comment and have updated the Data availability statement with accession numbers: “Data Availability: Shade-house experimental data are available in the NERC Environmental Information Data Centre at <https://doi.org/10.5285/f1d17e61-bb6c-47a9-a648-062c63ea7f16>. Sequence data have been deposited in the NCBI Sequence Read Archive (SRA) under Project ID PRJNA627300. Field census data are available upon reasonable request from the ForestGEO data portal at <http://ctfs.si.edu/datarequest/>.”

Reviewer #4 (Remarks to the Author):

This is a resubmission and I believe that Liang et al. satisfactorily addressed the reviewer’s comments. While this is an observational study and doesn’t provide direct causation, it does provide very compelling evidence that mycorrhizal networks are likely a possible mechanism in explaining forest species diversity. This is by no means the last study of this emerging paradigm and I hope this study will spur the scientific community to help advance this hypothesis into a possible theory.

Overall, this was a well-polished article and easily understandable. All my comments are minor, but I believe they will improve the readability.

Q20: Throughout the manuscript it is stated that high biodiversity is expected with AM-associated tree species due to a negative PSF (e.g. ln 44 & 203). While references

are provided, the mechanism is never really explained. Why would a high pathogen load can cause high diversity – what's the paradigm? You state "...which maintain high diversity by reducing the survivorship of conspecific seedlings in high density patches..." Based on this explanation, it still doesn't exclude the possibility of monodominance or a few genera taking over. It is just not obvious in the manuscript and I think 1-2 sentences can provide a fully-formed explanation.

Response: We agree with this comment and have added more detailed explanations on L45: *“Negative plant-soil feedback (PSF) effects mediated by soil-borne pathogens are widely detected in forest communities^{6,9-11}, which could cause disproportionately high mortality of conspecific^{10,11} and closely related¹² seedlings at high density near their parent trees, making resources available for distantly related species that are resistant to those natural enemies, helping to maintain high species diversity in the forests.”*

Q21: Figures 2,3,4 & Sup 1: Please change the bar graphs into box plots to provide data transparency and it is more informative. Overall, bar graphs basically hide most of the results by just showing average and SE, at least with a box plot the reader can get a sense of how the data was distributed within the treatment.

Response: We agree with this comment and have changed Figures S1 and S2 into box plots as suggested. However, in Figures 3 and 4, we were not showing average biomass and SE for each treatment but log-response ratios which compared seedling total biomass between in-growth cores with 35 μm or 0.5 μm mesh for each focal species at each site. We estimated the 95% confidence interval of each log-response ratio by bootstrap resampling with 9999 repetitions, which could not be presented with box plots. Also, in Figure 2 each treatment comprised of 48 binary 1/0 (survival/dead) data points, which is not applicable to display with dot or box plots as well. We have provided all of these original datasets with our final manuscript.